# Socialisation Agents’ Use(fulness) for Older Consumers Learning ICT

**DOI:** 10.3390/ijerph20031715

**Published:** 2023-01-17

**Authors:** Torgeir Aleti, Bernardo Figueiredo, Diane M. Martin, Mike Reid

**Affiliations:** School of Economics, Finance and Marketing, College of Business, RMIT University, P.O. Box 2476, Melbourne, VIC 3001, Australia

**Keywords:** consumer socialisation, socialisation agents, older consumers, ICT knowledge, older adults

## Abstract

This research investigates the socialisation agents older consumers use to learn about information and communication technologies (ICT). We surveyed 871 older consumers in Victoria, Australia, about whom they would most likely turn to for advice (i.e., their preferred socialisation agents) if they needed help using or fixing an ICT device. They were asked to identify the most and second most likely source of advice. Participants were also asked to assess the usefulness of the advice received from their preferred agents and to estimate their level of ICT knowledge. The findings reveal that older consumers tend to rely on younger family members. Still, the agency they receive from non-familial sources is essential when preparing for a digital consumer role. Surprisingly, ICT knowledge is determined by the socialisation agency received by older adults’ second advice option—which is less likely to be their own adult children. This research expands current knowledge about how older consumers perceive various ICT socialisation agents. Consumer socialisation theory suggests that socialisation agents impact how consumers function in the marketplace. Although the first choice of socialisation agent may be perceived as beneficial for older adults, the advice given does not relate to marketplace functioning regarding improved ICT knowledge.

## 1. Introduction

A generational ‘digital divide’ is highly problematic in our rapidly ageing world population [1]. Older adults, in particular, benefit from remaining active internet users, as doing so can be instrumental in maintaining social relationships and preventing loneliness [2]. However, technological changes have resulted in many older adults feeling disempowered and excluded from the marketplace, often fearing social exclusion if they lack sufficient digital literacy [3]. In addition, ageism and the stereotyping of older consumers as technologically disadvantaged are commonplace, influencing how technology is developed, marketed, and adopted—leaving older consumers on the low-use side of the technological divide [4,5]. This paper examines older consumers’ experiences with learning to use information and communication technologies (ICT) which has largely been neglected in extant consumer research.

A common stereotype about older consumers is that they stubbornly refuse to adopt technology and are ‘stuck in their ways’ [6,7]. Lee and Coughlin [8] suggest that slow adoption among older consumers may be related to an assumption that ICT innovations are for the young and that there are limited benefits of adoption by older consumers. Prensky [9,10] refers to the generational digital divide gap consisting of digital natives and digital immigrants. Older consumers are the immigrants who seek to immigrate to the land of the younger digital natives. Such ‘migration journeys’ are not well understood; who influences these journeys, and what are the goals of the immigrants? However, some research shows that older consumers are less anxious and more confident around ICT than this stereotype suggests [11,12]. Older consumers’ increased interest and willingness to adopt ICT have also increased due to physical isolation restrictions related to the COVID-19 pandemic [13,14].

Based on the theory of planned behaviour, a large body of literature has investigated ICT use and acceptance among older adults [15]. However, although social influence from social surroundings is known to influence older adults’ likelihood to use ICT [15], less focus has been placed on understanding how older adults obtain knowledge about ICT from their social surroundings.

Age-associated changes in cognition, affect, and goals interact to affect older consumers’ decision-making processes and habits, which differ from younger consumers [16]. Prior research has generated important insights concerning older consumers in cognitive domains, but contextual factors that influence these behaviours are not well understood [17]. This is surprising since consumers are influenced by their background, experiences, and learning accumulated through the life course [18,19]. Furthermore, consumer experiences over one’s lifetime are also influenced by social contexts, which shape one’s socialisation process [20,21,22]. This paper adopts a consumer socialisation perspective to extend our understanding of how older adults learn about ICT and prepare for a digital consumer role.

Consumers obtain relevant marketplace skills and knowledge through socialisation processes. Consumer socialisation is the learning process related to the marketplace and is defined as acquiring skills, knowledge, and attitudes to function in the marketplace [23]; in short, the process of learning all aspects of the consumer role [24]. As such, understanding the socialisation process is central to consumption studies since all consumption is learned [25]. Furthermore, it is widely acknowledged that consumer socialisation, like all forms of socialisation, is a life-long process as consumers continue to update and adjust skills and knowledge throughout their lives [19,23]. However, the ways older consumers update skills and knowledge to address marketplace changes, including engagement with ICT, has been under-investigated and poorly understood [24,25].

Socialisation theory suggests that an individual’s skills, knowledge, and attitudes are constantly influenced by social surroundings, including people (e.g., family, peers) and social institutions (e.g., schools, religion). These socialisation agents shape skills, knowledge, and attitudes [23,26]. When the marketplace changes, older consumers need to re-learn aspects of consumer behaviour to continue functioning in the marketplace [27,28]. Older consumers may turn to socialisation agents based on self-interest and social influence from these agents. For instance, social influence is a significant factor driving older consumers toward online shopping [29]. Younger family members are also central socialisation agents for older consumers’ ICT and are often relied on for decision-making and learning new ICT-related skills [30].

The purpose of this study is to gain a better understanding of the ways socialisation agents shape older consumers’ learning and engagement with ICT. This paper investigates the ICT socialisation agents, including family members and others, who influence older consumers to understand how different agents are engaged and what socialisation outcomes these agents provide to older consumers.

## 2. Theory and Research Questions

### 2.1. Socialisation as Role Preparation

A fundamental purpose of socialisation is ‘role preparation’ [26]. Role preparation relates to the anticipation of new roles in, for example, occupations or institutions such as marriage and parenthood [31]. Through socialisation, people acquire skills and attitudes compatible with their roles [26] (e.g., parent and grandparent roles). People gradually change their identity to fit the assumed or anticipated role and engage in activities consistent with it [32]. Behaviours at a given stage in life often relate to accumulated experiences (e.g., behaviours, socialisation) at earlier stages [33], making late-life socialisation dependent on all lived experiences [19]. Indeed, older consumers need to consolidate decades of consumer experiences to weigh the benefits and trade-offs with new modes of consumption [34,35].

Preparing for a new consumer role includes anticipating needs by re-learning, updating, or adjusting consumption skills [36]. However, role preparation has only been indirectly addressed by consumer research. In the context of older consumers and ICT, this may include using search engines to research purchase decisions, online banking, online shopping/selling, or making appointments and bookings. In addition, such role preparation means developing a level of digital literacy that enables ICT consumption behaviours, underpinned by basic technical use of computers and the internet [37]. Late-life consumer role preparations around ICT relate to taking on the role of a digitally enabled consumer—the ‘digital consumer’.

Many older consumers seek to migrate towards an increasingly digitalised society [9,10] by adjusting their consumer behaviour in ways that add to their already established consumption patterns. Older consumers are designated digital migrants, while younger consumers are digital natives [9,10]. However, digital natives unfamiliar with digital immigrants’ past consumption patterns and preferences may not understand older consumers role-preparation goals towards becoming digital consumers [30]. Although it is widely acknowledged that younger family members often act as socialisation agents when assisting older consumers with ICT, it is unclear how older consumers use and perceive other sources of knowledge. These additional sources of knowledge, including salespeople or peers, are also socialisation agents [26].

### 2.2. Socialisation Agents: Family, Friends, and Others

Early research has established that much adult socialisation is self-initiated and voluntary [38], e.g., individuals may choose friends based on mutual interests [39,40]. These individuals have more resources and alternatives, allowing withdrawal from an agent if the socialisation process is not proceeding as expected [40]. Senior consumers purposely seek socialisation agents that fulfil their consumption needs based on the individual consumer’s perception of their own needs.

Socialisation agents are acknowledged as fundamental to consumer learning in childhood and earlier stages of adulthood [41]. Socialisation agents transmit norms, attitudes, and values to learners by actively or passively communicating certain expectations and behavioural patterns [23]. It is widely recognised that family, peers, and the media are essential socialisation agents early in life [26,42]. It is further acknowledged that children obtain different consumer knowledge from these socialisation agents [43]. However, little focus has been placed on understanding how older consumers perceive support from different socialisation agents and the relationship between older adults’ ICT knowledge and the socialisation support they receive.

Family members may play a key role in technology acceptance by older consumers. Research has investigated the role of younger family members as essential socialisation agents for older consumers [44,45]. Luijkx et al. [46] argue that the influence of each family member has its own characteristics when it comes to teaching ICT to older family members. They argue for including all family members when investigating ICT in the lives of older consumers. While their adult children may focus on use(fulness) concerns, grandchildren tend to focus on entertainment and pleasure, which grandparents often appreciate. Xiong and Zuo [47] further argue that emotional support from family members has a more substantial impact on improved internet literacy than cognitive support. However, the socialisation process within families is fuelled by tension, frustration, and accusations from all parties involved; the old are unable/unwilling to learn, and the young are unable/unwilling to teach [30,48]. Perez et al. [49] notes that the family is central to socialisation as ageing parents depend on their adult children for ICT expertise.

Although younger family members are regarded as essential socialisation agents for ICT learning for older consumers, the assumption that the young act as socialisation agents for the old seems to challenge established family roles of parents being the socialisation agent responsible for educating their children [30,48]. Despite this, older parents tend to favour their adult children as socialisation agents since non-familial socialisation agents, such as professional sources, are seen by older adults as lacking credibility in providing helpful marketplace information [22]. Furthermore, Selwyn [50] argues that although adult children (and grandchildren) may appear to be a significant official socialisation agent for older adults’ ICT usage, they are rarely the sole reason for their parents’ interest in ICT. Older adults often compare their ICT skills with peer groups, another important source for ICT interest [51] and the consequent socialisation process towards preparing for the digital consumer role.

The general strength and usefulness of a socialisation agent’s advice is measured as Consumer Socialisation Agency (CSA) [52]. CSA has been investigated from a family perspective, assessing the level of agreement between agents and learners [53]. CSA provides insights into consumer socialisation processes in dyadic relationships between closely related agents and learners [54]. However, research in CSA has primarily focused on the socialization agent [55] as opposed to the learner. Research has investigated what motivates someone to offer socialisation agency, but not how learners perceive receiving agency from different agents. Therefore, our research questions are:RQ1Who are the principal socialisation agents for older consumers’ role preparation as digital consumers?RQ2What levels of socialisation agency do older consumers perceive they receive from different agents?RQ3How does consumer socialisation agency from different agents influence ICT knowledge attainment?

## 3. Materials and Methods

Survey data in this study focused on ICT use among older consumers. Participants were encouraged to identify their socialisation agents when preparing for the digital consumer role. Voluntary participants, members of a non-profit senior organisation, were asked about the ICT socialisation agents preferred and the perceived usefulness of the preferred agents. The survey was distributed to the 4000 members of the organisation through their online and paper-based newsletters. A total of 871 responses were collected (630 paper, 241 online). Most consumers surveyed were female (78%), and 90% were above age 65.

Participants were asked to identify whom they would most likely turn to for advice (i.e., their preferred socialisation agents) if they needed help using or fixing an ICT device. They were asked to identify the most and second most likely source of advice from a list of the most common socialisation agents identified by previous research (i.e., adult child, grandchild, partner, a friend of the same age, younger friend, professional, sibling, or describe another source if none of the options suited their situation).

Participants were also asked to assess the usefulness of preferred agents using Aleti Watne et al.’s [52] Consumer Socialisation Agency (CSA) scale. The CSA scale measures the level of acceptance of an agent’s influence by the learner about consumption [52]. The CSA measures the extent to which the learner accepts and absorbs skills, knowledge, and attitudes from a specific socialisation agent within a particular consumption context. The CSA measures five items on a 7-point scale (anchored in “Strongly disagree” and “Strongly agree”). Item example: “I get useful information from my [#1] about technology devices”, “I feel more confident about using technology devices when guided by my [#1]”, and “I use the technology devices my [#2] suggests”. For each of the items in the CSA scale, participants would replace #1 and #2 with the first and second most preferred socialisation agent they had previously indicated in the survey. We limited responses to the two most common socialisation agents to avoid complexity in the survey instrument, which could lead to confusion or fatigue in participants. The scale measured CSA from the agents participants actually used.

The survey instrument measured ICT knowledge using Bloch, Ridgway, and Sherell’s [56] simple subjective measure of product knowledge. Participants were asked to rate their level of ICT knowledge on a 7-point scale (anchored in “little or no knowledge” and “a great deal of knowledge”). The survey covered the broad common product categories of computers, smartphones, and tablets.

## 4. Results

In response to RQ1, the most frequently used socialisation agents were relatives. Almost half of the participants first turned to their adult children for socialisation agency. Table 1 summarises the preferred socialisation agents by category. Some participants defined their socialisation agents outside of the survey options. These self-initiated options were most frequently the internet (Google or YouTube). Some participants referred to formal or semi-formal ICT education, such as computer classes for older adults or the library.

Younger family members and friends were by far the most reported category of preferred socialisation agents. In total, 57.3% had their first choice of socialisation agent in this category, and 50.2% used this category as their second option (Table 1). Although adult children were the most likely agent to be used, they were not perceived as the most helpful socialisation agents (Table 2). This is surprising, given the overwhelming preference for ICT help from one’s own adult children. The finding suggests a complex consumer socialisation relationship between adult children and their older parents. Friends and family members of similar age were the second most frequently used group of socialisation agents. In this category, 23.8% had their first choice of socialisation agents, and 25.6% had their second option (Table 1). Moreover, this category of socialisation agents provides significantly less agency than younger family members and friends (Table 2). Although often turned to, these ‘fellow digital migrants’ are of less value as socialisation agents for older adults when preparing for the digital consumer role.

In response to RQ2, Table 2 shows differences in how valuable participants found the agency from their first and second options of socialisation agents. A one-way ANOVA between the first option socialisation agent categories demonstrated a strong statistically significant difference in perceptions of CSA (F = 5.04, Sig. 0.00), with self-initiated agents being the most useful and siblings the least useful. Although there were no statistically significant differences between the second option (F = 1.59, Sig. 0.14), the second option was seen as significantly less helpful than the first option overall. Moreover, agents were less valuable for most individual agency categories when they were the second option instead of the first.

Service professionals were also an essential source of consumer socialisation agency for older adults preparing for the digital consumer role. For this category, 15.2% of the survey participants reported having service professionals as their first option, and 21.8% had them as their second option. Although professional help is often sought, older adults perceive their advice as less valuable (Table 2). Notably, the view of the CSA provided by service professionals did not differ significantly between the first and second options. The agency from a service professional is viewed similarly whether it is the first or second option for advice. Finally, 3.8% used the ‘other’ option to describe their most important agent, and 2.4% used it for their second agent (Table 1). Despite the low number of participants preferring these agents as their first and second option, they provided the highest level of consumer socialisation agency (Table 2).

To address RQ3, we conducted a regression analysis using Bloch et al.’s [56] measure of self-perceived level of ICT knowledge as the dependent variable (i.e., the consumer socialisation outcome) and consumer socialisation agency from the first and second preferred agent as independent variables (Table 3). Even though the findings presented in Table 2 suggest that the first preference socialisation agent gives better agency, this agency does not lead to ICT knowledge outcomes. Surprisingly, all regressions confirm a significant relationship with CSA from the second option of socialisation agents and no significant relationships for the first option of agents. Thus, more than one agency source is needed for older adults when preparing for the digital consumer role as it is the second—not the first—source that aligns with their perceived ICT knowledge.

## 5. Discussion

Older consumers often rely on the ICT expertise of family members and service professionals. They balance this reliance with other sources of influence and evaluations of need based on a myriad of accumulated consumption experiences throughout their life. Extant literature on consumer socialisation suggests that the family is a vital socialisation agent for children and young adults; it is evident that the family continues to be an essential socialisation agent throughout the lifecycle. Although this is potentially not surprising, older consumers do not always see the CSA from family members as the most valuable when preparing for the digital consumer role. This is particularly the case when family members are from the same generation (i.e., spouse/partner or sibling).

Across adulthood, individuals become increasingly responsible for their own socialisation [32]. When learning about ICT and preparing for the role of digital consumers, older consumers seem to value the socialisation agency received the most when self-initiated. Participants who chose the ‘other’ option in the survey and added self-initiated options, such as the internet or computer classes for older adults, reported the highest level of consumer socialisation agency. However, using the internet as a socialisation agent requires a certain level of digital literacy (i.e., using Google to search for information). Consequently, other socialisation agents need to be engaged to reach this level of ICT competency. This reinforces socialisation as a life-long process that aims to prepare for and perform an increasingly complicated consumer role.

Consumer socialisation theory refers to how socialisation agents affect the mental and behavioural characteristics of the learner; they do so by providing knowledge and information or building values and norms to follow [57]. The first choice of socialisation agent for older consumers learning about ICT does not consistently impact the learners’ mental and behavioural characteristics. Although the first choice of socialisation agents may indeed provide sought-after knowledge and information, the second choice of agency relates to the knowledge level.

This research was conducted in Australia, so the findings may only be applicable to that context. In Australia and other Western countries, adult children seldom live with their parents. This makes socialisation and ICT help within families challenging, as it often involves several households, geographic boundaries, and limited availability of help [58,59]. Consequently, the findings here focused on intergenerational socialisation agency, and agent preferences may relate to views of autonomy, independence, and family relations within the research context. Other ICT-socialisation studies, such as Perez et al.’s [49] in Mexico and Aleti et al.’s [53] in Vietnam, revealed complex role negotiations and levels of authority within co-habiting multigenerational families. Future research should investigate socialisation agency in different countries and compare the results.

## 6. Conclusions

The results suggest that older consumers rely more on their first choice of socialisation agents than their second choice. However, their choices are also subject to agent availability. The significant differences in consumer socialisation agency received between the first and second option of agents suggest a drop in perceived agency received from the second option. Despite this, the agency received from the second option of socialisation agents, as opposed to the first, related to participants’ level of ICT knowledge. A possible interpretation of this counterintuitive finding is that for some, the first agent may provide them with all the knowledge they need. This confirms previous research suggesting that adult children often complain about repeatedly assisting their parents with the same ICT problems [30,48]. Thus, we suggest that older consumers sometimes outsource the performance of the digital consumer role to the first socialisation agent and postpone their own role preparation. Adult children often act as ICT influencers in a family context, and their parents model their consumption after them [49]. Instead of acquiring skills, knowledge, and attitudes about new technology from their children, ageing parents depend on their children’s ICT expertise [49]. This may be related to perceptions of risk associated with ICT use [60]. However, they are less likely to look for outsourcing as their second choice because they receive this from their first choice, so knowledge and agency will align.

Almost half of the participants indicated adult children as the most frequently used socialisation agent for ICT. Furthermore, adult children were the second-highest valued agent in terms of the CSA provided. However, the perception of socialisation agency from adult children drops significantly when they are the second option. Younger friends are seen as the most valuable second-option agents, providing the highest CSA. Keeping in mind the CSA provided from the second option of agents related to ICT knowledge in our regression, a variety of agents are needed to assist older consumers in preparing for the digital consumer role. The first option of agents may indeed provide ICT-related socialisation agency, but this agency does not relate to continuous role preparation. Future research should investigate how different socialisation agents contribute to older consumers’ ongoing or postponed role preparation.

The result of this study indicates that some older consumers self-socialise [61,62] by utilising the internet as a socialisation agent. As such, the internet needs to be seen as a socialisation agent and compared with the other agents. Emerging research on consumer socialisation has started to investigate this, for example, by perceiving user-generated content on social media as a socialisation agent [57]. Future research should focus on conceptualising how the internet functions as a non-human agent of self-socialisation with other human socialisation agents.

## Figures and Tables

**Table 1 ijerph-20-01715-t001:** Preferred 1st and 2nd options of socialisation agents.

Consumer Socialisation Agent	% 1st Option	% 2nd Option
Younger family members and friends	57.3	50.2
Adult child	46.8	22.9
Grandchild	6.6	19.2
Younger friends	3.9	8.1
Same-age family members and friends	23.8	25.6
Spouse/partner	14.1	7.0
Friends same age	7.1	14.5
Sibling	2.6	4.1
Professional		
Service professionals	15.2	21.8
Self-initiated		
Internet, library, computer class	3.8	2.4

**Table 2 ijerph-20-01715-t002:** Relative strength of CSA received from categories of agents.

Socialisation Agent Category	Mean (Std. Dev.) CSA #1	Mean (Std. Dev.) CSA #2	Diff. between #1 and #2 (sig)
Younger family members and friends			
Adult child	5.22 (1.28)	4.38 (1.42)	−0.84 (<0.00) *
Grandchild	4.99 (1.38)	4.26 (1.65)	−0.73 (<0.00) *
Younger friends	4.80 (1.53)	4.55 (1.40)	−0.25 (0.22)
Same-age family members and friends			
Spouse/partner	4.71 (1.32)	4.30 (1.52)	−0.41 (0.09)
Friends same age	4.50 (1.29)	4.13 (1.32)	−0.37 (0.01) *
Sibling	4.32 (1.62)	3.54 (1.43)	−0.78 (0.01) *
Professional			
Service professionals	4.53 (1.74)	4.47 (1.64)	−0.06 (0.70)
Self-initiated			
Internet, library, computer class	5.48 (1.70)	4.51 (1.61)	−0.97 (0.05) *
Mean	4.95 (1.44)	4.32 (1.50)	−0.63 (<0.00) *
**ANOVA between categories (sig.)**	**F = 5.04 (<0.00)**	**F = 1.59 (0.14)**	

CSA #1 = CSA from first preferred option of socialisation agent. CSA #2 = CSA from second preferred option of socialisation agent. * = sig.

**Table 3 ijerph-20-01715-t003:** Linear regression: ICT knowledge and CSA from first and second agents.

Dependent Variable		CSA 1st Agent Option			CSA 2nd Agent Option		
	R²	β	T	P	β	T	P
Knowledge Computers	0.03	0.07	1.53	0.126	0.12	2.73	0.007 *
Knowledge Smartphones	0.05	0.09	1.71	0.088	0.19	3.96	<0.001 *
Knowledge Tablets	0.03	0.10	1.56	0.120	0.17	3.04	0.002 *

* = sig.

## Data Availability

The data presented in this study are available on request from the corresponding author. The data are not publicly available due to privacy and ethical restrictions imposed by the Research Ethics Board.

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
