# Peer review of "Socialisation Agents’ Use(fulness) for Older Consumers Learning ICT"

_ijerph, 2023, doi:10.3390/ijerph20031715_

Round 1
Reviewer 1 Report
Dear Author(s),
The present manuscript has offered an investigation of the socialisation agents and their perceived usefulness to receiving older consumers. This is a timely topic in an increasing digital world and important work that can inform the bridging of well-theorised digital divides.
The manuscript is in great shape. It has a clear and straightforward story. The presentation of the findings is easy to follow and offers insights that are in the sweet spot of slightly unexpected but indeed make sense when explained. The discussion of role preparation is informative and I agree that this perspective needs development in consumer research. I also appreciated the authors defusing common stereotypes regarding older consumers which enables a fair analytical treatment of this quite heterogeneous group of individuals.
My comments are nothing major but offer suggestions to tighten the manuscript further. These concern a couple points to further clarify, as well as a couple of considerations that are worth acknowledging regarding older consumer tech consumption that can further contextualise the results in ways that can inform readers.
Quick Clarification Points:
· Considering the wide variety of sources of socialisation listed, what is the rationale of analysing two options? Why not analysing the third or beyond? Is this for simplicity, methodological reasons, or both? This offers a chance to show more of the rationale behind research design choices.
· Page 7, line 314: What does older consumers ‘outsourcing’ the performance of the digital performance of the digital consumer role to the first socialisation agent and postpone their own role preparation entail? What does this look like in practice? This is quite abstract and hard to follow, so a sentence or two will help illustrate and clarify here.
Some further considerations:
1: Co-location or physical separation of older consumers and their adult children. Prior consumer research does point out the challenges of older consumers no longer living with their adult children which can make socialisation/tech-help challenging or at least its dynamics shifting. This might help your explanations of some participants’ preferences and perceptions for technology assistance in relation to your point about agent availability. Two references that are helpful here:
· Marchant, C., & O'Donohoe, S. (2014). Edging out of the Nest: Emerging Adults' Use of Smartphones in Maintaining and Transforming Family Relationships. Journal of Marketing Management, 30(15-16), 1554-1576.
· Franco, P. (2020). Empowering the Independence of Older People with Everyday Technologies. In S.A. Churchill, L. Farrell & S. Appau (Eds.), Measuring, Understanding and Improving Wellbeing Among Older People (pp. 15-39). Singapore: Palgrave MacMillan.
2: Professional sources of help can come in the form of services that visit homes or require consumers to take devices to retail locations (e.g., Apple Genius Bar). These can also range in the extent of services and their costs, and some might be perceived better as fixing problems than providing coaching/lessons, and vice-versa. Are these distinctions considered in the survey design? How might participants be thinking about this question?
Overall, thank you for your paper, it was a pleasure to review. As such, I hope that the suggestions I have provided above are helpful and I wish you all the best on your journey to successfully publish this manuscript.
Reviewer 2 Report
The article presents interesting research results on how older people acquire digital skills.
The presented research results allow to achieve the aim of the work. The results are described correctly.
There is little space in the article for discussion. It is worth supplementing the article with the results of research presented in other countries.
Reviewer 3 Report
To support the reviewed manuscript, I am sending specific comments:
1. This paper investigates the ICT socialisation agents, including family members and others, who influence older consumers.
The main research questions are:
- Who are the principal agents of socialisation for older consumers’ role preparation as digital consumers?
- What levels of socialisation agency do older consumers perceive they receive from different agents?
- How does consumer socialisation agency from different agents influence ICT knowledge attainment?
2. The topic described in the manuscript is very important and requires constant research. Although it is widely acknowledged that younger family members often act as socialisation agents when assisting older consumers with ICT, it is unclear how older consumers use and perceive other sources of knowledge.
3. Compared to other publications, the research provides a new theoretical tool to understand how different agents are engaged and what socialisation outcomes these agents provide to older consumers.
4. The research methodology is clear, understandable and very well presented. The reviewer has no comments. The research objectives are correct and the research results are clearly presented.
5. The conclusions are consistent with the evidence and arguments and relate to the intended purpose of the study.
6. The references used in the manuscript are appropriate and current. 59 references were used.
The reviewer's recommendation to continue research in other countries and to compare the results.
Considering the above, I recommend the article to be published in the IJERPH.
